# What Happens in Male Dogs after Treatment with a 4.7 mg Deslorelin Implant? I. Flare up and Downregulation

**DOI:** 10.3390/ani12182379

**Published:** 2022-09-12

**Authors:** Sabrina Stempel, Hanna Körber, Larena Reifarth, Gerhard Schuler, Sandra Goericke-Pesch

**Affiliations:** 1Reproductive Unit—Clinic for Small Animals, University of Veterinary Medicine Hanover, Foundation, 30559 Hannover, Germany; 2Clinic for Obstetrics, Gynecology and Andrology of Large and Small Animals, Klinikum Veterinärmedizin, Justus-Liebig-University Giessen, 35392 Giessen, Germany

**Keywords:** deslorelin, GnRH stimulation test, hCG stimulation test, male dogs, flare up

## Abstract

**Simple Summary:**

Until now, information about the “flare up” and the time to downregulation in male dogs after treatment with a 4.7 mg deslorelin implant is strongly limited, regarding testosterone concentrations, testicular and prostatic volume and semen quality. The aim of this study was to provide detailed insights into these open questions. GnRH and hCG stimulation tests were performed to gain further insights into testicular endocrine function. Seven male beagle dogs were treated with a 4.7 mg deslorelin implant, and three animals were treated with saline, representing the controls. In deslorelin-treated dogs, first basal testosterone concentrations were observed earliest on D7 and latest on D28 after treatment. Infertility—based on the lack of semen or spermatozoa— was diagnosed earliest on D35 and latest on D77. After five months, the treatment was still effective in six dogs but was reversed in one deslorelin-treated dog.

**Abstract:**

Although registered since 2007, knowledge about changes in testosterone concentrations (T), testicular and prostatic volumes (TV, PV) and semen quality, as well as the time point of infertility following treatment with a 4.7 mg deslorelin (DES) slow-release implant, is limited. Therefore, seven sexually mature male dogs were treated with DES (TG); three male dogs treated with saline served as controls (CG). The study assessed local tolerance, TV, PV, semen parameters and T subsequent to GnRH/hCG stimulation in regular intervals. Local tolerance was good. In TG, T was increased right after treatment, but decreased four hours afterwards. Subsequently, TV, PV, semen quality and T decreased over time in TG, but not CG. T was basal (≤0.1 ng/mL) from D28 onwards. Response to GnRH/hCG stimulation was variable, with two TG dogs having increased T post-stimulation on all study days independent of pre-treatment concentrations. A(zoo)spermia in TG was observed from D35–D77 in all seven dogs. Whereas treatment was still effective in six TG dogs five months after implant insertion, it was fully reversed in one dog in terms of T and spermatozoa on the last examination. These results indicate high variation in individual dogs, necessary to consider when advising dog owners.

## 1. Introduction

Interest in alternatives to surgical castration has increased worldwide in the last years [1,2,3,4,5,6,7,8,9,10]. Possible reasons are legal aspects, with elective neutering (without medical indication) being illegal in countries such as Norway and Germany [11,12], ethical and personal (also regional) preferences, dislike of the irreversibility of the surgical option [2] and fear of short- and mid-term side effects (risk of anesthesia, risk of surgery) [13,14], but also increased owners’ awareness about possible long-term consequences of surgical castration [13,15], such as obesity, certain orthopedic problems or tumor predispositions [16,17,18,19,20,21]. Slow-release GnRH agonist implants (SRI) have been extensively investigated and are considered a suitable reversible alternative to surgical castration [2,5,6,7,8,10,22,23,24,25,26]. Application of a SRI results in an initial stimulation of pituitary gonadotropin and testicular testosterone secretion, the so-called “flare up”, followed by downregulation of endocrine and germinative testicular function, resulting in a reversible castration effect including basal testosterone concentrations and infertility [10]. Vickery et al. were the first to describe the use of GnRH agonists for chronic contraception in dogs [27,28]. Afterwards, various GnRH agonist depot formulations were described in the literature to successfully suppress testicular function in male dogs, namely buserelin [6,29], azagly-nafarelin [10] and deslorelin [7,8,22,23,24,30,31,32,33].

The only GnRH agonist implant currently approved in Europe and marketed for induction of temporary infertility in healthy adult, intact, sexually mature male dogs is Suprelorin^®^, containing 4.7 or 9.4 mg deslorelin (Virbac, Carros, France). Although it has been widely used since its introduction to the market in 2007 and extensive clinical experience is available, the effects on germinative and endocrine testicular function associated with its application have been poorly characterized. Only a small number of studies investigated the effects of a Suprelorin^®^ 4.7 mg deslorelin implant on prostatic size [30,33] and on semen quality [7,30]. More detailed information on the type and dynamics of deslorelin-induced changes in testicular functions should be considered very helpful for use in daily clinical practice, and it is important to fully advise owners of male dogs who choose deslorelin SRI as a medical alternative to castration. Therefore, the aim of this study was to closely monitor the effect of the 4.7 mg deslorelin implant on pituitary–testicular axis function and germinative testicular function over a period of five months after application.

## 2. Materials and Methods

Animal experimentation approval was given by the corresponding authorities (AZ 19/3203, LAVES Hannover).

### 2.1. Animals and Experimental Design

Ten mature male beagle dogs, 2.5 years of age with a mean body weight of 12.5 ± 2.5 kg (range: 10–15 kg), were included. All males were trained and used to the procedure of semen collection before the beginning of the study. They were housed in different groups, in kennels with doghouses at the university facilities. The experimental study design is shown in Figure 1. A randomized controlled blinded laboratory study design was used. Prior to treatment with a deslorelin implant or saline, dogs were clinically and andrologically examined including semen collection and evaluation. In addition, ultrasound examinations of the prostate gland were performed and blood samples for testosterone analysis were taken. Additionally, a GnRH stimulation test using buserelin acetate (0.4 µg/kg Receptal^®^ IV, Intervet Deutschland GmbH, Unterschleißheim, Germany) and a hCG stimulation test using human chorionic gonadotropin (750 IU/dog IM, Ovogest^®^, Intervet Deutschland GmbH, Unterschleißheim, Germany) on the subsequent day were performed to investigate the hypothalamus–pituitary–gonadal axis (buserelin) and Leydig cell function (hCG) prior to treatment. On day 0, the day of treatment, animals were randomly divided into two groups, a treated group (TG, n = 7) receiving a 4.7 mg deslorelin implant (Suprelorin^®^, Virbac, Carros, France) and a control group (CG, n = 3) being injected with 0.5 mL sterile saline. All animals were treated subcutaneously into the paraumbilical area following clipping and disinfection of the injection area. Whereas the senior author (S.G.-P) performed the treatments, the examiner (the first author) was blinded to the groups when performing all clinical examinations, semen collections/evaluations and blood samplings. Animals were followed in detail over five months with the implant in place. The last examination day for semen collection, ultrasound of prostate gland and GnRH (hCG) stimulation tests was D147, and hCG stimulation was performed on D148. Additionally, a single blood sample for testosterone analysis and an andrological examination were also performed on D154. After five months (154 days), the right testes of all treated and control animals and the implants (TG only) were removed (D154 corresponds to D ex).

### 2.2. Physical and Andrological Examination including Evaluation of Implantation Site

Physical examinations, including body weight and clinical parameters, such as rectal temperature, heart rate and breathing frequency, were conducted regularly over the course of the study, namely daily from D0 (treatment) until D7 and afterwards every two weeks until the end of the study. Additionally, injection sites were examined daily for swelling, reddening, pain and heat between D1 and 7.

Andrological examinations, including measurements of testicular dimensions and recording of testicular consistency, were performed weekly by the same blinded person (first author) to avoid operator-related variance [33]. Testicular dimensions were obtained using a sliding caliper as previously described [34,35]. The testicular volume (cm³, TV) was calculated by the formula of an ellipsoid (length × width × height × 0.5236) [34].

### 2.3. Ultrasound of the Prostate Gland

The size of the prostate gland was measured weekly for five months by ultrasound (LOGIQ S7 Expert Vet, Scil animal care company GmbH, Viernheim, Germany) using a convex scanner probe (10 C, 10.0 MHz). Dogs were examined in standing position. The prostate gland was scanned sagittally for measuring length and height, and transversally for height and width [36,37]. Measurements of sagittal and transverse images of the prostatic length, height and width were performed in duplicates for calculating mean values and to calculate prostatic volume (PV) using the formula for an ellipsoid [length × (height + height/2) × width × 0.523)] [36,38,39].

### 2.4. Semen Collection and Analysis, Urine Examination

Semen collection was attempted/performed on a weekly basis during the whole course of the study, namely from prior to treatment to five months afterwards. Semen was collected fractionatedly into glasses by manual stimulation in the presence of a teaser bitch [40]. Semen collection was stopped when no further secretions were obtained or when erection ceased, but at latest after 10 min. The teaser bitch was either in estrus or an additional cloth with vaginal discharge from an estrus bitch was used for stimulation. Libido was evaluated subjectively. Regarding semen analysis, each fraction was examined for volume (VOL, mL) and microscopic presence of spermatozoa. Detailed microscopical examination was performed for the “sperm-rich” fraction including—if sufficient volume available—estimation of the percentage of progressively motile spermatozoa (% PM) of viable (% VS) and morphologically abnormal spermatozoa (% MAS). Furthermore, the total sperm count (×10^6^, TSC) was calculated based on the sperm concentration (using a Thoma chamber) and the volume of the respective ejaculate fraction. For determination of the percentage of progressively motile spermatozoa (% PM), three individual droplets of the “sperm-rich” fraction were prepared on a pre-warmed glass slide, covered with coverslips, and PM was estimated at 160× magnification using a phase contrast microscope with a heating stage (38 °C; Zeiss/Winkel 135133, Göttingen, Germany). For viability (VS), an eosin-stained sperm smear (Sigma-Aldrich Chemie GmbH, Munich, Germany) was prepared and 200 spermatozoa were evaluated (if possible) [41] at 400× magnification using bright field microscopy (Zeiss West, Oberkochen, Germany). Stained sperm cells indicate damaged sperm membranes, whereas unstained sperm cells indicate intact sperm membranes. For MAS, a semen aliquot was fixed with 300 µL formol citrate (2.9 g trisodiumcitrate dihydrate and 100 mL double distilled water, 4 mL of the solution replaced by 4 mL of 37% formaldehyde solution; Carl Roth GmbH + Co. KG, Karlsruhe, Germany) [35,37]. A droplet of the fixed semen was placed on a glass slide, covered with a coverslip and 200 sperm—if enough sperm were available—were evaluated for their morphology at 1000× magnification (oil immersion) using a phase contrast microscope (Zeiss West, Oberkochen, Germany) [42]. In the case of azoospermia (absence of spermatozoa in the ejaculate) or aspermia (no ejaculate can be collected), a spontaneous urine sample was taken right after the semen collection and centrifuged for 10 min at 600× *g* to microscopically examine the sediment for the presence of spermatozoa at 400× magnification using phase contrast (Zeiss West, Oberkochen, Germany). Dogs were considered infertile in the case of absence of spermatozoa in the urine at three subsequent occasions. 

### 2.5. Blood Collection, GnRH/hCG Stimulation Tests and Hormone Analysis

Blood (2 mL plasma) for testosterone analysis was collected by venipuncture (Vena cephalica antebrachii or Vena saphena) into lithium–heparin-coated tubes prior to treatment, on D0 (1 h, 4 h, 10 h, 24 h), D7, 14, 28, 42, 56, 84, 112 and 154 after treatment. Blood samplings prior to treatment and on D0 1 h, D7, etc., were performed in the morning at the same time point on each study day. Samples were centrifuged at the earliest 10 min and maximum 1 h after sampling for 10 min at 1892× *g* using a Heraeus Megafuge 2.0 R (Landgraf Laborsysteme HLL GmbH, Langenhagen, Germany). Plasma was stored at −80 °C after aliquoting until analysis. For analyzing plasma testosterone concentrations, a well-described and validated RIA [43,44] was used with a lower detection limit of 0.05 ng/mL. The intra- and interassay coefficients of variation were 3.7% and 7.6%, respectively. Additionally to single blood samples for testosterone analysis, GnRH/hCG stimulation tests were performed on specific occasions, namely prior to treatment and on D7/8, 28/29, 56/57, 84/85, 112/113 and D147/148 after implant/placebo injection. On the first mentioned day, the GnRH stimulation test was performed, followed by the hCG stimulation test on the subsequent day at the same time point. On these days, intravenous catheters (B. Braun Vet Care GmBH, Tuttlingen, Germany) were placed in the Vena cephalica antebrachii and used for blood sampling into lithium–heparin-coated tubes. For both stimulation tests, an initial blood sample (pre-treatment) for testosterone was taken using an IV catheter. For the GnRH stimulation test, buserelin acetate (Receptal^®^; 0.4 µg/kg, Intervet Deutschland GmbH, Unterschleißheim, Germany) was injected intravenously right after blood collection [9], and another blood sample was taken one hour after injection of the GnRH agonist treatment and saline flushing using an IV catheter (post-treatment). For the hCG stimulation test, 750 IU hCG/dog (Ovogest^®^; Intervet Deutschland GmbH, Unterschleißheim, Germany) was injected intramuscularly right after the initial blood collection; 60 min later the second sample was taken using an IV catheter (post-treatment) [9,45]. Centrifugation and preparation of blood samples were performed as described above. The stimulation tests were performed to confirm a functional hypothalamus–pituitary–gonadal axis (GnRH agonist buserelin) or to test for the functionality of Leydig cells (hCG) over time, with an increase in peripheral testosterone concentrations in the second sample (post-treatment) confirming a functional physiological hypothalamus–pituitary–gonadal axis and functional Leydig cells. 

### 2.6. Hemicastration and Implant Removal

All surgeries were performed by the senior author, unblinded to the grouping. Animals were hemicastrated five months (154 days) following treatment with the implant or placebo. Orchiectomy of the right testis was performed under general anesthesia with propofol 2–4 mg/kg (CP-Pharma, Burgdorf, Germany) and ketamine 1 mg/kg (CP-Pharma, Burgdorf, Germany), after premedication with dexmedetomidine 5 µg/kg (CP-Pharma, Burgdorf, Germany) and levomethadone 0.2 mg/kg (MSD, München, Germany) maintained by isoflurane (1%) and following general surgical principles. The testes were immediately processed within the next minutes after surgery. After removal of the respective testis, the paraumbilical area was surgically prepared (scrubbed, disinfected) at the location of the implant. Following preparation, a skin incision was made where the implant could be palpated, and the implant was carefully surgically removed in total. In the case that the implant broke, specific care was taken to ascertain complete removal of all pieces. Afterwards, the previous implantation site was flushed with sterile saline. The incision was closed with a skin suture using Vicryl 3/0 (Ethicon, Norderstedt, Germany). All dogs received 4.0 mg/kg carprofen (Rimadyl^®^, Zoetis Deutschland GmbH, Berlin, Germany) for three days.

### 2.7. Processing of Testicular Tissues and Further Evaluation

Tissue samples were fixed in Bouin’s solution for 24 h and subsequently washed several times with 70% ethanol every 24 h. After embedding in paraffin, samples were cut using a microtome (Leica Microsystems, Wetzlar, Germany), stained with hematoxylin eosin, and thereafter mounted with Roti^®^ Histokitt (Carl Roth GmbH + Co. KG, Karlsruhe, Germany). Evaluation was performed using 400× magnification (Zeiss West, Oberkochen, Germany) to evaluate the most developed germ cells and the level of arrest of spermatogenesis, as previously described [26]. To identify and specify Sertoli cells and spermatogonia, immunhistochemistry stainings were performed against Vimentin and DAZL (deleted in azoospermia-like), respectively. 

The aim was to compare staining intensity and quantity of stained cells in testis specimen of both deslorelin- and saline-treated animals. Sections were cut and immunhistochemistry (IHC) was performed as previously described [46,47]. The testis sections were deparaffinized and rehydrated. After antigen retrieval by boiling in 0.01 M sodium citrate buffer (pH 6), sections were rinsed in distilled water and the tissue was treated with 3% H_2_O_2_ to block endogenous peroxidase activity. For blocking of non-specific protein binding sites, Vimentin slides were incubated with 10% horse serum (S-2000, Vector Laboratories, Burlingham, CA, USA) in 5% bovine serum albumin (BSA, VWR Life Science, Solon, OH, USA) and DAZL slides with 10% goat serum in 3% BSA at room temperature for 30 min. Subsequently, slides were incubated with the monoclonal mouse anti-Vimentin clone V9 antibody (M072529-2, Dako, Agilent, Santa Clara, US, 0.78 × 10^−3^ µg/µL) or the polyclonal rabbit anti mouse antibody (ab34139, Abcam, Cambridge, UK, dilution 2 × 10^−3^ µg/µL) overnight at 4 °C. After washing with ICC (1.2 g Na_2_HPO_4_, 0.2 g KH_2_PO_4_, 0.2 g KCl, 8.0 g NaCl, 3 mL Triton ad 1000 mL), slides were incubated for 30 min with the appropriate secondary antibody (Vimentin: horse anti-mouse—BA-2000, Vector Laboratories; DAZL: goat anti-rabbit—BA-1000, Vector Laboratories). Immunoreactivity was visualized by an immunoperoxidase method according to the manufacturer’s instructions (VECTASTAIN PK-6100 Rabbit IgG Elite ABC Kit and Vector Nova-RED Substrate Kit SK-4800, Vector Laboratories). Negative (ICC buffer only) and isotype controls (Vimentin: irrelevant mouse IgG, I-2000, Mouse IgG, DAZL: irrelevant rabbit IgG (I-1000, Rabbit IgG; both: Control Antibody, Vector Laboratories) were included in the run. All slides were counterstained with Mayer’s hematoxylin, dehydrated and mounted with Roti^®^ Histokitt (Carl Roth GmbH + Co. KG, Karlsruhe, Germany). For histological examination of Vimentin and DAZL, immunopositive signals were evaluated descriptively using an Olympus BX41TF Microscope (Olympus^®^, Shinjuku, Japan) with an Olympus DP72 camera (Olympus Corporation, Tokio, Japan) and the Olympus cell Sense Dimension Software (version 2.1, Olympus Corporation, Tokio, Japan).

### 2.8. Statistical Analyses

Statistical analysis was performed after the end of the complete study period, with the first author being unblinded at this time point. 

Microsoft Excel (2019, Microsoft Corporation, Washington, DC, USA) was used for data management and graphical representations; statistical analysis was performed using GraphPad prism (GraphPad Prism Software 8.3.0, San Diego, CA, USA). The aim of this study was to identify differences between TG and CG and within the respective group over time. In general, data of all dogs were used for statistical analysis. However, to evaluate changes in semen parameters due to treatment, only dogs with normospermia [42] prior to treatment (n = 6) were included in the statistical analysis. 

To determine whether data were normally distributed, the Shapiro–Wilk test was applied. For comparative reasons, all data regarding TV, PV, total ejaculate volume (VOL), PM, VS, MAS and TSC were presented as median and first/third quartile (Q1/Q3). To identify significant differences over time within one group (TG/CG), a repeated measures ANOVA was performed in the case of normal distribution (CG: VOL, PM, VS, MAS, TV and PV). For not normally distributed data, the Friedman test (non-parametric, paired) was applied (CG: TSC, testosterone; TG: VOL, TSC, TV, PV, testosterone). To compare “prior to treatment values” of both groups, the parametric unpaired t-test was used in case of normal distribution (TV, VOL, PM, VS, TSC, testosterone). If data were not normally distributed, non-parametric unpaired test was used (Mann–Whitney test; MAS, PV). Results with *p* ≤ 0.05 were considered statistically significant. In addition to raw data, the TV and the PV were given as relative % volumes considering the TV/PV of day 0 as 100% and the respective days as relative percentage results. Due to the low ejaculate volumes and/or low sperm numbers on some collection dates during effective treatment, not all ejaculate parameters could be assessed on all study days. Full data sets were only available for VOL and TSC. Consequently, the other parameters were only described. For the graphical presentation, all available information was used, even if the number of animals was low. Testosterone concentrations were given as geometric mean and dispersion factor [$\bar{x_{g}}$ (DF)]. For further statistical analysis, values below the detection limit of the RIA (<0.05 ng/mL) were defined as 0.025 ng/mL. Testosterone concentrations of ≤0.1 ng/mL were considered as basal [10,25,48]. In addition, when GnRH/hCG stimulation tests were performed, the concentration before stimulation (pre-stimulation) was considered as 100% and the after stimulation testosterone concentration (post-stimulation) was given as relative change to the pre-stimulation sample (times increase to before). The relative changes in testosterone concentrations over time were compared statistically by the applying Friedman test. 

## 3. Results

### 3.1. Physical Examinations and Evaluation of the Treatment Site

All dogs were clinically healthy before the start of the study and stayed healthy throughout the whole observation period. The treatment site was asymptomatic in all animals of CG (n = 3) and two dogs of TG (n = 2). Neither swelling, reddening, pain nor heat were observed in any of these dogs. In the remaining dogs of TG, the treatment site was reddened and swollen in two dogs on the first day. Whereas one of these did not show any further signs, in the second dog the implantation site was swollen until D2 and reddened until D5. Additionally, one dog showed reddening only on D1, another dog on D2 and 3 and one on D3 and 4, respectively. Symptoms in all dogs, however, were low-grade at any time and neither pain nor heat occurred. 

### 3.2. Andrological Examination

Consistency of the testes was firm-elastic in CG at all examination time points but changed from firm-elastic to soft-elastic in all treated dogs between D35 and D56.

Median TV in CG varied between 9.6 cm^3^ (Q1: 9.5 cm^3^; Q3: 10.9 cm^3^) prior to the start of the study and 10.6 cm^3^ (Q1: 9.7 cm^3^; Q3: 11.0 cm^3^) five months after treatment. In TG, total TV was a median of 8.7 cm^3^ (Q1: 8.0 cm^3^; Q3: 9.1 cm^3^) prior to treatment, reached the nadir on D105 with 2.9 cm^3^ (Q1: 2.6 cm^3^; Q3: 3.3 cm^3^) and was 3.1 cm^3^ (Q1: 2.8 cm^3^; Q3: 3.4 cm^3^) five months after treatment.

Prior to the start of the experiment, absolute TV of both groups did not differ significantly between groups (*p* = 0.1867). Whereas absolute TV (cm^3^) of CG did not differ significantly over time (*p* = 0.1782), absolute TV of TG was significantly affected by time (*p* < 0.0001), with an obvious reduction from D21 onwards. Changes in the relative TV over time are given in Figure 2. As expected, the relative TV differed significantly over time in TG (*p* < 0.0001). Although the Friedman test revealed significant differences in the relative TV over time in CG (*p* = 0.0124), subsequent multiple comparisons failed to verify individual differences. The relative TV was reduced by about 60% on D56 in TG and remained at approximately this volume until five months after treatment, the end of the initial observation period (Figure 2). Comparing TG and CG per time point, statistical analysis revealed significant differences regarding the TV from D28 onwards.

### 3.3. Ultrasound of the Prostate Gland

The absolute PV of CG animals varied in median between 11.3 cm^3^ (Q1: 11.0 cm^3^; Q3: 11.6 cm^3^) prior to the start of the study and 9.7 cm^3^ (Q1: 8.9 cm^3^; Q3: 9.8 cm^3^) on D147. In TG, the total median PV was 9.9 cm^3^ (Q1: 8.9 cm^3^; Q3: 13.7 cm^3^) prior to treatment, decreased to 2.1 cm^3^ (Q1: 2.0 cm^3^; Q3: 2.7 cm^3^) on D70 and reached the nadir of 1.8 cm^3^ (Q1: 1.7 cm^3^; Q3: 2.2 cm^3^) on D147. Absolute PV of both groups did not differ significantly (*p* = 0.8333) prior to treatment. While no effect of time was observed in CG (*p* = 0.2141), the prostatic size in TG was significantly affected by time (*p* < 0.0001). Changes in the relative PV (% PV) over time are given in Figure 3. Different from the relative PV in CG, showing no significant differences (*p* = 0.1057), the relative PV of TG was significantly reduced over time (*p* < 0.0001), with the volume being approximately 26% of the initial size on D63 and only 18% on D147. Values of PV of TG were significantly different compared to CG from D42 onwards. Furthermore, echogenic appearance changed from a homogenous organ structure with a “butterfly shape” to a more heterogeneous structure with hypoechogenic parts and a loss of the “butterfly shape” in TG.

### 3.4. Semen Collection and Analysis

Prior to treatment, semen quality was in the normal range [42] in 9/10 dogs. Semen quality was reduced at various occasions prior to the start of the study and during pre-treatment examination in one dog without any obvious reason (Murphy). As only dogs with normospermic ejaculates were included in the statistical analysis, spermatological results of this dog are also presented in the individual courses of treated animals (Figure 4, Figure 5, Figure 6, Figure 7 and Figure 8).

All investigated semen parameters, total ejaculate volume (mL, VOL), percentage of progressive motility (PM) of viable (VS) and morphologically abnormal spermatozoa (MAS) and total sperm count (TSC), were not different between CG (n = 3) and TG (n = 6) during the pre-treatment examination prior to the start of the study (VOL, PM, VS, TSC: *p* = 0.4540–0.8120; MAS: *p* = 0.2976). Furthermore, VOL (*p* = 0.1350), PM (*p* = 0.3789), VS (*p* = 0.4084) and MAS (*p* = 0.1834) did not change significantly in CG over time. In contrast to this, TSC was affected over time in CG (*p* = 0.0263). 

Due to low ejaculate VOL and low sperm concentration in specific samples of TG, not all ejaculate parameters could be analyzed on all days. Consequently, except for total ejaculate VOL and TSC, the remaining semen parameters are presented descriptively only. In general, semen parameters varied considerably over time in TG. Whereas VOL, PM, VS and TSC decreased over time, MAS increased during the study period. Median VOL in TG decreased from D14 onwards and reached its lowest values at D49 and remained at this level (Figure 4a,b). Median PM decreased until D28/35 (Figure 5a,b). The higher median PM values on D42/49 were related to the fact that ejaculates could only be collected from two dogs. For the graphical presentation all available data were used, even if the number of animals was low (but the number of animals is indicated). The percentage of VS also decreased over time from D21 onwards, but median values (≥83%) remained still in the reference range for normospermia (Figure 6a,b). Different from this, the percentage of MAS increased from D7 to nearly 100% from D35 on (Figure 7a,b). Median TSC decreased from D21 onwards and reached a nadir on D49, remaining low until the end of the study period (Figure 8). Statistical analysis revealed that VOL (*p* < 0.0001) and TSC (*p* < 0.0001) were significantly affected by treatment and time in TG. 

Considering individual dogs, azoospermia (absence of spermatozoa in the ejaculate) was reached (TSC = 0) on D35 (n = 1), D42 (n = 3), D49 (n = 1), D56 (n = 1) and D77 (n = 1). Urine was devoid of sperm for the first time on D42 (n = 2), D49 (n = 3), D63 (n = 1) and D84 (n = 1), indicating complete absence of sperm. Interestingly, azoospermic ejaculates could be collected in four of six animals during the remaining observation period, whereas in two dogs no ejaculate could be collected (aspermia). Even though libido was significantly reduced in all treated animals, it was affected most in aspermic animals of TG. Three dogs of TG (with azoospermic ejaculates) showed pelvic thrusts during the weeks of downregulation.

Of note, spermatozoa were present in the sample of one dog of TG on D147 (PM: 20%, MAS: 97%), whereas all others were still a(zoo)spermic.

Considering the data of the dog not included in the statistical analysis (Murphy), prior to treatment, semen quality was reduced at various occasions and during pre-treatment examination in this dog (Murphy) without any obvious reason. Spermatological examinations showed oligozoospermia and teratozoospermia on most of the pre-treatment days. Right after treatment, the total ejaculate volume started to decrease rapidly from D7 to volumes less than 0.5 mL from D28 onwards (Figure 4b). The percentage of PM increased from the day of treatment to D7, followed by a decrease afterwards. On D28, no progressively motile spermatozoa were found (Figure 5b). Prior to treatment the percentage of VS was 92%, starting to decrease from D21 to 38.5% on D28 (Figure 6b). The percentage of MAS increased from D14 onwards, peaking on D21. Afterwards, values were decreased on D28 (Figure 7b). The TSC was increased on D14 compared to prior to treatment values but decreased afterwards with azoospermia reached on D35 (Figure 8b). Urine did not contain any spermatozoa from D42 onwards. Despite differences in the initial semen samples, observed semen trends identified in TG dogs and the reduction in libido became also obvious in Murphy, however, with a more dramatic decrease in VS compared to the remaining dogs.

### 3.5. Testosterone Concentrations and Stimulation Tests

Independent of group, basal testosterone concentrations prior to the start of the study varied between 0.34 ng/mL and 7.84 ng/mL ($\bar{x_{g}}$ (DF); CG: 1.67 (4.07) ng/mL; TG: 4.17 (2.10) ng/mL) and did not significantly differ between TG and CG (*p* = 0.2049). Subsequent to implant treatment, testosterone concentrations increased initially (D0 +1 h), but started to decrease already four hours after treatment, reaching basal testosterone concentrations (≤0.1 ng/mL) earliest on D7 in one dog and latest on D28 in the other six dogs (Figure 9). Testosterone was >0.1 ng/mL in two dogs on D112 and in five of seven dogs on D ex. 

Comparing TG and CG, TG testosterone concentrations were significantly different from D28 onwards and remained significantly lower compared to CG until D112. However, they no longer differed between groups on D147 and D ex.

Testosterone concentrations in TG differed significantly over the study period (*p* < 0.0001), whereas they did not differ over time in CG (*p* = 0.5796). 

GnRH stimulation tests (using buserelin) were performed prior to treatment and on D7, 28, 56, 112 and D147 after treatment to test for a functional hypothalamus–pituitary–gonadal axis. The relative increase in testosterone concentrations post-GnRH stimulation did not differ between groups prior to deslorelin/saline treatment. Concentrations one hour subsequent to stimulation were increased in contrast to pre-stimulation values in both CG and TG (Figure 10). Prior to the beginning of the study, one dog in TG showed no testosterone increase post-stimulation with buserelin (T_pre_ = 4.8 ng/mL; T_post_ = 4.7 ng/mL). On D7, another dog of TG had basal testosterone (<0.1 ng/mL) pre- and post-GnRH stimulation. Similarly, on D28 all dogs in TG (seven out of seven) had basal pre-stimulation testosterone, but only two out of seven dogs in TG had basal concentrations post-stimulation, consequently not responding to stimulation. Whereas all pre-stimulation testosterone concentrations were <0.1 ng/mL on D56 in TG, two of seven dogs still had a positive response with testosterone concentrations >0.1 ng/mL post-stimulation. The same observation was made on D84 in these two and another dog (three/six: Lui, Toni, Fiete; T_post_ > 0.1 ng/mL). On D112 four of seven dogs and on D147 (five months) five of seven dogs had basal testosterone pre-stimulation. On both dates (D112 and D147), four out of seven (D112: Lui, Toni, Fiete, Murphy; D147: Lui, Toni, Spencer, Murphy) showed a positive response (T_post_ > 0.1 ng/mL) to GnRH agonist stimulation. To summarize, two out of six dogs (Lui/Toni) responded with a testosterone increase to buserelin injection at all sampling time points, even when revealing basal pre-stimulation testosterone concentrations. Related to this, the results of TG did not differ significantly over time (*p* = 0.1687), just as values of CG (*p* = 0.1682).

HCG stimulation tests were performed prior to treatment and on D8, 29, 57, 113 and 148 after treatment to test for Leydig cell functionality. Before deslorelin/saline treatment, the increase in testosterone concentrations of both groups post-hCG stimulation did not differ. In CG, one dog (Knopf) showed no testosterone increase after hCG administration prior to the start of the study, but on D8, 29, 57, 113 and 148. Another dog of CG (Merlin) showed a characteristic increase in testosterone subsequent to stimulation at the beginning of the study, but testosterone did not significantly increase post-hCG stimulation on D57, 85 and 113 compared to pre-stimulation concentrations. Testosterone concentrations in TG increased in five out of seven dogs as a consequence to hCG treatment prior to deslorelin treatment and in seven out of seven on D8. On D29, D57, D85 and D113, testosterone was <0.1 ng/mL in all animals of TG pre-hCG stimulation, but concentrations increased >0.1 ng/mL in three of seven animals (Lui, Toni, Strolch) post-stimulation (Figure 11). Furthermore, on D148 all pre-stimulation blood samples revealed testosterone values <0.1 ng/mL and increased >0.1 ng/mL in two of seven dogs (Lui, Toni) post-stimulation. Therefore, to summarize, two of seven dogs responded to hCG stimulation on all study days when hCG stimulation tests were performed (Lui, Toni). Noteworthy, the same dogs responded to GnRH stimulation with a testosterone increase. Relative testosterone increases in TG differed over time (*p* = 0.0023), while those of CG were not significantly affected by time (*p* = 0.1476).

### 3.6. Histology of Testes

Spermatogenesis differed considerably between dogs. While all CG and one TG dog had normal spermatogenesis with fully elongated spermatids present five months after treatment, spermatogenesis was arrested at the level of spermatogonia in four of the treated dogs and at the level of spermatocytes in the remaining treated dog (Table 1, Figure 12). Subjectively, Leydig cell nuclei seemed to be reduced in size in dogs with downregulated spermatogenesis compared to control dogs. Different to this, Leydig cell nuclei size did not obviously differ between the control dogs and the treated dog where spermatogenesis was already restarted.

#### 3.6.1. Vimentin Protein Expression

IHC revealed specific immunopositive staining against Vimentin located in the cytoplasm of Sertoli cells in the seminiferous tubules in all groups (Figure 13). Sertoli cell nuclei were round to ovoid and frequently detached from the basal membrane in four of six deslorelin-treated dogs, concomitant with spermatogenesis arrested at the level of spermatogonia. Different to this, Sertoli cell nuclei were polygonal and positioned close to the basal membrane in dogs with full spermatogenesis, independent of saline and reversed deslorelin treatment (n = 1).

#### 3.6.2. DAZL Protein Expression

IHC revealed specific immunopositive staining against DAZL located in the nuclei of spermatogonia and early spermatocytes in all groups (Figure 14). Subjectively, the number of immunopositive signals in spermatogonia was reduced in testicular tissues of animals, showing an arrest of spermatogenesis at the level of spermatogonia and/or spermatocytes compared to testicular tissues obtained from saline-treated controls. In addition, the staining intensity seemed to be reduced in some tubules in downregulated canine testes compared to controls. In contrast, neither the number of immunopositive signals nor the staining intensity differed considerably between the deslorelin-treated dog that had reversed and CG, indicating reversibility of induced effects (Figure 14).

## 4. Discussion

### 4.1. Evaluation of the Implantation Site and Implant Removal

Implantation sites of five out of seven deslorelin-treated dogs showed mild, transient local reactions that disappeared within three days in all but one dog, where the reddening remained up to five days after treatment. The good local tolerance is in agreement with previous observations of the 4.7 mg deslorelin implant in dogs [30] and cats [49,50,51,52]. All implants could be easily removed during general anesthesia necessary for hemicastration, as described earlier in anesthetized cats [49].

### 4.2. Testicular and Prostatic Volume

Due to the loss of testicular endocrine and germinative function, testis volume was significantly reduced during treatment and testicular consistency was changed from firm-elastic to soft-elastic in TG, whereas it was not affected by time in CG. The reduction in TV by about 60% on D56 (week 8) in seven out of seven dogs in our study is in good agreement with earlier observations reporting a maximum of 65% reduction after a 3, 6 or 12 mg deslorelin SRI [31] or a 4.7 mg deslorelin SRI [33], but slightly less than the 75% reduction reported after 6 mg deslorelin SRI [22]. Similarly, the change in testicular consistency was also already described before with other implants (azagly-nafarelin, buserelin) [6,10,29]. 

Similarly, PV was significantly reduced over time, namely by 73% on D63 (week 9) and by 82% on D147 (week 21) in TG in our current study. Even though the reduction in volume and size is well known and was shown earlier with different GnRH SRI, our study is the first to describe the PV reduction in healthy animals over time following the administration of a 4.7 mg deslorelin SRI. Only two studies have investigated PV reduction following application of a 4.7 mg deslorelin SRI, however, in dogs with benign prostatic hyperplasia. The maximum reduction in PV was by 66% in various breeds [33] and 67% in German Shepherds [30], reported 24 weeks and 52 days after treatment, respectively. 

### 4.3. Testosterone

The significant reduction in TV and PV is a consequence predominantly of downregulation of testicular endocrine function. In general, physiological testosterone concentrations in intact male dogs range between 0.4 to 6.0 ng/mL [53], which is in good agreement with the pre-treatment concentrations in both groups. 

A temporary increase in gonadotropins and testosterone is reported to occur following administration of GnRH SRI due to their initial stimulatory effect. The so-called “flare up effect” is frequently feared by pet owners and veterinarians due to a potential worsening of, e.g., behavioral problems or symptoms of prostatic disease. By now, detailed data about the duration of the “flare up” are currently only available for the 6 mg deslorelin SRI, with an increase being observed 40 min, a peak at one hour and a return to pre-treatment testosterone five hours after treatment [22]. The present study sheds some light on the short-term effect of a 4.7 mg deslorelin SRI on peripheral testosterone concentrations by regularly repeated samplings and allows for the conclusion that the “flare up” is short-term, with a testosterone increase observed one hour after treatment, but a decrease in concentrations not significantly different from those prior to treatment four hours after treatment.

Long-term treatment results in downregulation of testicular function, resulting in basal testosterone concentrations. According to our previous publications, testosterone concentrations ≤0.1 ng/mL are considered basal, indicating efficacy of treatment [10,25,48,54]. However, unfortunately, there is no consensus about the definition of “basal” in the literature. Some studies investigating the effect of deslorelin SRI consider concentrations <0.1 ng/mL [30], <0.4 ng/mL [7] or <1 ng/mL [55] basal, making comparison between different studies difficult and the terminology “castration-like” even questionable according to the before mentioned testosterone reference ranges and diurnal variation of intact male dogs. Considering earlier studies using a 4.7 mg deslorelin SRI, testosterone was not detectable from day 11 onwards [30], or decreased significantly between day 14 and 21 after treatment [56]. In our study, testosterone was basal earliest on D7 (week 1) (one dog) and latest on D28 (week 4) in six out of seven dogs. The fact that one dog already had testosterone >0.1 ng/mL on D147 (week 21) and normal spermatogenesis (and semen) indicates that in some animals the duration of effect can be shorter than the manufacturer-postulated six months. Interestingly, five of seven dogs no longer had “basal” testosterone concentrations after five months. It remains to be clarified whether the high frequency of the short duration of effect in our study (predicted to act six months by the manufacturer) is by coincidence, a general observation/issue or related to regularly performed GnRH and hCG stimulation tests during the observational period.

As testosterone is secreted in a pulsatile pattern, resulting in considerable diurnal variation [53,57,58], single individual testosterone measurements are not considered optimum [57,59]. To provide deeper insights into the testicular endocrine function, we performed GnRH and hCG stimulation tests. The aim of the GnRH stimulation test using the GnRH agonist buserelin was to investigate whether the (hypothalamus–)pituitary–gonadal axis is (still) functional. During effective treatment of GnRH SRI, internalization and downregulation of pituitary GnRH receptors had been postulated as part of the mode of action [60,61], indicating that testosterone concentrations will not be increased subsequent to injection of GnRH agonists. On the other hand, hCG stimulation tests aimed to confirm functionality of Leydig cells by responding with increased testosterone production subsequent to stimulation of their LH receptors. Using an azagly-nafarelin SRI, we showed that at the time of basal testosterone concentrations the steroidogenic capacity was reduced due to reduced mRNA and protein expression of the steroidogenic acute regulatory (StAR) protein and steroidogenic enzymes [48,62]. Our study is, to the best of our knowledge, the first assessing the functionality of the pituitary–gonadal axis and Leydig cells following treatment with a 4.7 mg deslorelin SRI in male dogs. Administration of native GnRH in male dogs, treated with a 6 mg deslorelin implant, had been investigated earlier [23], just as GnRH [51] and hCG stimulation tests [52,63] in tom cats following application of 4.7 mg deslorelin SRI, and confirmed to be suitable to provide insights into the mode of action and onset of efficacy. Considering the observed effects in our study, it is noteworthy that a discrepancy between the time point of basal single/individual testosterone concentrations (only one sample taken or pre-stimulation sample) and the time point of basal testosterone concentrations following stimulation was observed. Whereas basal individual testosterone concentrations were reached after 28 days (week 4) in all dogs without stimulation (see above), testosterone was first basal in five of seven dogs on D56 (week 8) post-GnRH stimulation. Different to this, concentrations of the post-GnRH stimulation were not basal in the remaining two dogs even at all later time points following GnRH stimulation. In contrast, testosterone responses following administration of native GnRH were significantly reduced 15 days after implantation of a 6 mg deslorelin implant, and dogs did not even respond to stimulation at the end of the observation period, namely 100 days after implant treatment [23]. Additionally, an earlier study described four out of five cats already reaching basal testosterone concentrations pre- and post-GnRH stimulation four weeks after implant application [51]. 

The increase in testosterone in three out of seven dogs subsequent to hCG stimulation (post-stimulation) on various occasions (D29, 57, 85 and 113) in spite of basal pre-stimulation testosterone indicates functional Leydig cells—at least in some of the treated animals. Whereas no further testosterone increase was observed following hCG in cats treated with a 4.7 mg deslorelin SRI [63] and subsequent to bovine LH in dogs treated with 6 mg deslorelin SRI [23], results similar to ours were observed in another “cat study” using the 4.7 mg deslorelin SRI [52].

Comparing the relative testosterone increase subsequent to buserelin and hCG, it became obvious that hCG induced a bigger increase than the GnRH agonist in TG animals, possibly pointing to an (more) affected hypothalamus–pituitary–gonadal axis or an altered expression of GnRH receptors, as postulated earlier [64]. However, the fact that there is still an increase in testosterone, even if it is slight, indicates that downregulation of pituitary GnRH receptors is extremely variable. Different to this, the relative increase in testosterone subsequent to buserelin and hCG was similar in CG. Interestingly, in our current study some dogs showed a tendency to higher testosterone concentrations on days with GnRH stimulation than on the subsequent day with hCG stimulation. Whether this is coincidence or due to a specific reason, such as immunization, requires further investigations. Immunization and/or reduced response due to antibody formation might, however, be an explanation why one dog in CG did not show any response to hCG application from D57 to D113, as described after repeated use in mares during breeding season [65,66], cows [67], bulls [68], queens [69,70] and dogs [71].

### 4.4. Semen Evaluation

Whereas the slight variation in semen quality, namely total sperm count, in CG was likely related to a variable degree of sexual arousal (presence of an estrous teaser bitch or a bitch + pheromone-containing cloth) [72], the significant variation in TG was considered to be treatment-associated. It seems noteworthy that although the time to endocrine downregulation was relatively predictable and quick (28 days), downregulation of germinative function was significantly later and more variable. Interestingly, we did not see a temporary improvement of semen quality, as reported by Romagnoli et al. [7], but it is without a doubt that dogs have to be considered as potentially fertile initially after treatment. The observed decrease in ejaculate volume, PM and TSC had been reported earlier following deslorelin 4.7 mg SRI [7,30] and deslorelin 6 mg SRI [22], but also after 6.6 mg buserelin SRI [6] and 18.5 mg azagly-nafarelin [10]. Similarly, the identified significant increase in MAS was seen in some studies following deslorelin 4.7 mg SRI [30], deslorelin 6 mg SRI [22] and 6.6 mg buserelin SRI [6], but not in others [7].

Infertility subsequent to a 4.7 mg deslorelin SRI was reported after 35 days in six German Shepherds with semen collection being no longer possible [30], after 23 to 84 days in six dogs due to a(zoo)spermia (two out of six dogs) or grossly reduced semen quality (semen volume <0.5 mL; progressively motile spermatozoa <10^7^; four of six dogs) [7] and after 14 (n = 4) to 21 days (n = 1) when semen collection was no longer possible [56]. Different to our study, however, animals were not examined on a weekly basis in earlier studies. Still, onset of infertility, namely azoospermia, was observed from day 35 to 77 (5–11 weeks), similar to Romagnoli et al. [7], with animals being, however, truly infertile as confirmed by absence of spermatozoa in the urine. The fact that azoospermic ejaculates could be collected in some animals during the treatment period was described earlier subsequent to GnRH SRI [10] and surgical castration [72].

### 4.5. Histology and Immunhistochemistry

Histological assessment of testis revealed an arrest of spermatogenesis at the level of spermatogonia and/or spermatocytes in animals with low testosterone concentrations. In addition, tubular diameters were reduced, Sertoli cell nuclei were—at least partially—detached from the basal membrane, as confirmed by Vimentin staining, and Leydig cell nuclei seemed to be diminished in size, possibly pointing to reduced steroidogenic capacity. These observations are in good agreement with the changes in semen quality (see above) and the appearance of the “downregulated testis” subsequent to 6 mg deslorelin [24], 6.6 mg buserelin [6] and 18.5 mg azagly-nafarelin [26,48,62]. DAZL is a germ-cell-specific marker that is expressed by undifferentiated and differentiated spermatogonia as well as early spermatocytes in dogs [73]. Subjectively, the number of DAZL immunopositive cells was reduced in downregulated testis, confirming a significantly affected spermatogenesis by deslorelin SRI treatment in TG. Further studies are required to better elucidate the effect of GnRH agonist SRI implants on the spermatogonial stem cells and their niche. Different to this, spermatogenesis, Sertoli and Leydig cell appearance, but also DAZL staining, did not differ from CG in the dog of TG with high testosterone concentrations on the day of hemicastration, indicating reversal and reversibility of treatment. In addition, histology confirms the end of efficacy in the respective dog, indicating full reversibility, but also a highly variable duration of effect in a cohort of dogs.

## 5. Conclusions

This study confirms the suitability of a 4.7 mg deslorelin-containing SRI implant as a reversible alternative to surgical castration with a variable onset, but also duration of treatment (efficacy). Libido, including pelvic thrusts during semen collection, was maintained in some dogs even at the time of low or basal testosterone concentrations, although it was reduced compared to untreated intact dogs. Semen quality was significantly reduced during treatment, but time point and duration of infertility exhibited a considerable variation ranging from day 35 to day 84, with some dogs being already fully reversed (and consequently fertile) within the predicted duration of effect, namely six months. Consequently, veterinary practitioners and pet owners co-housing treated male dogs with intact females should be well aware of the individual variability and the fact that fertile matings might occur within six months after treatment with a 4.7 mg deslorelin SRI. Further (clinical) studies should confirm our findings on a large cohort of dogs of various breeds and body weights.

## Figures and Tables

**Figure 1 animals-12-02379-f001:**
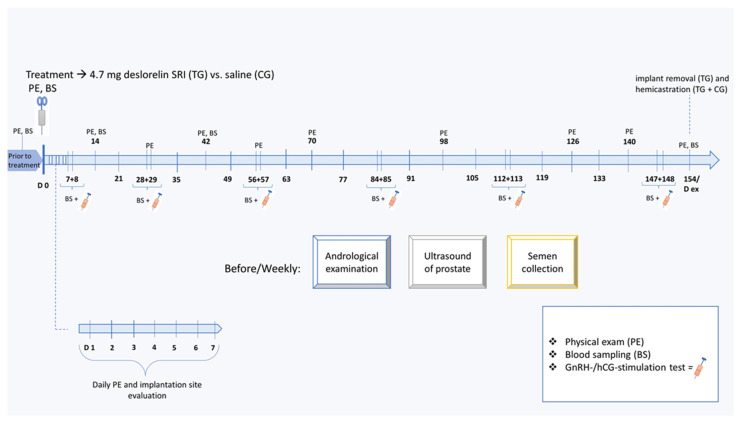
Detailed experimental design. The study includes two groups: TG animals (n = 7) were treated with a 4.7 mg deslorelin slow-release implant (SRI) into the paraumbilical area where CG animals (n = 3) were subcutaneously treated with 0.5 mL saline into the umbilical area.

**Figure 2 animals-12-02379-f002:**
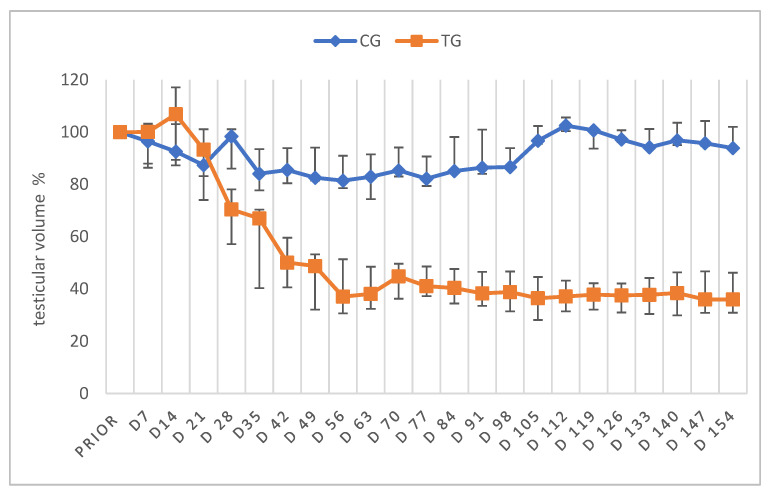
Testicular volume measured by a caliper given in % as median (Q1, Q3) over time; animals treated with a 4.7 mg deslorelin implant (TG, n = 7); saline-treated control animals (CG, n = 3). “Prior” indicates “prior to treatment”.

**Figure 3 animals-12-02379-f003:**
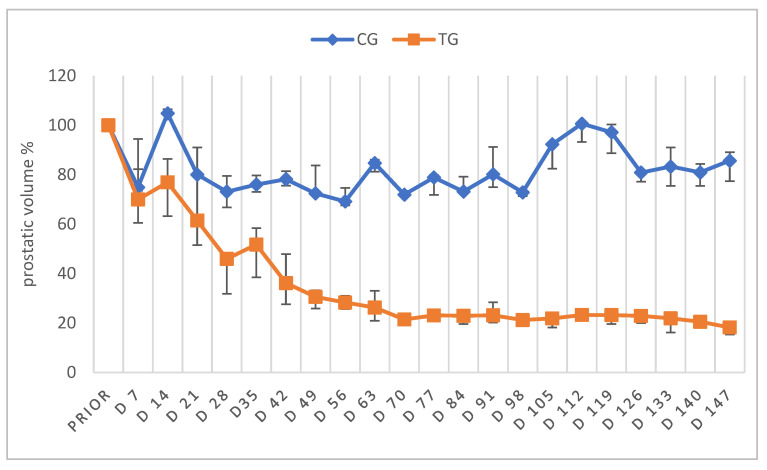
Prostatic volume given in % as median (Q1, Q3) over time of animals treated with a 4.7 mg deslorelin implant (TG, n = 7) and saline-treated control animals (CG, n = 3). “Prior” indicates “prior to treatment”.

**Figure 4 animals-12-02379-f004:**
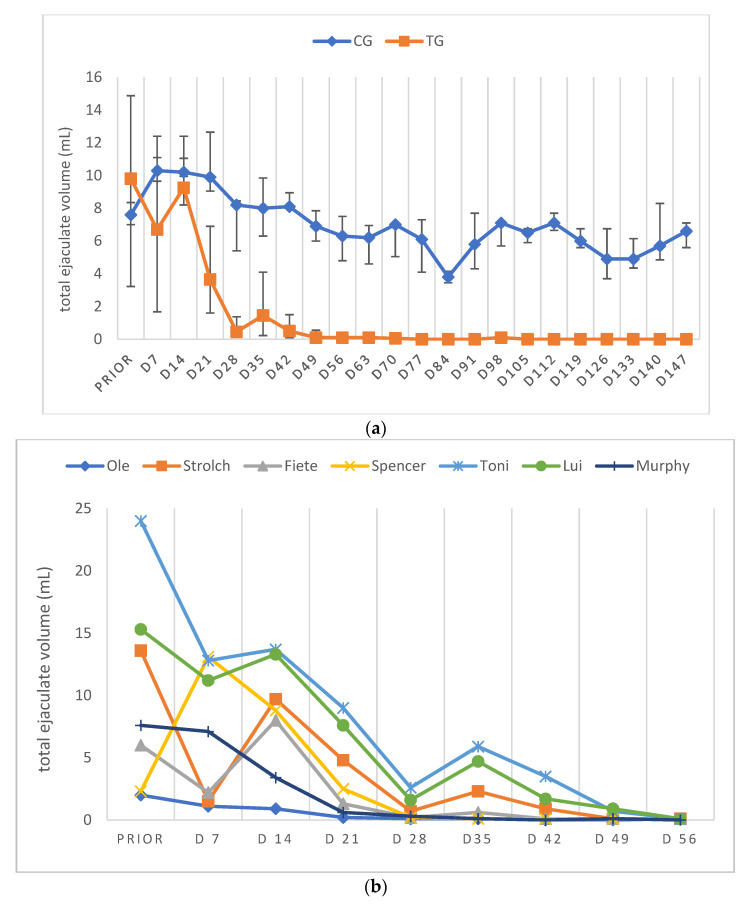
(**a**) Total ejaculate volume (mL) given over time (prior to treatment to D147) as median (Q1, Q3) of animals treated with a 4.7 mg deslorelin implant (TG, n = 6) and saline-treated control animals (CG, n = 3). “Prior” indicates “prior to treatment”. (**b**) Total ejaculate volume (mL) given as individual courses of animals treated with a 4.7 mg deslorelin implant (TG, n = 7) over time. “Prior” indicates “prior to treatment”.

**Figure 5 animals-12-02379-f005:**
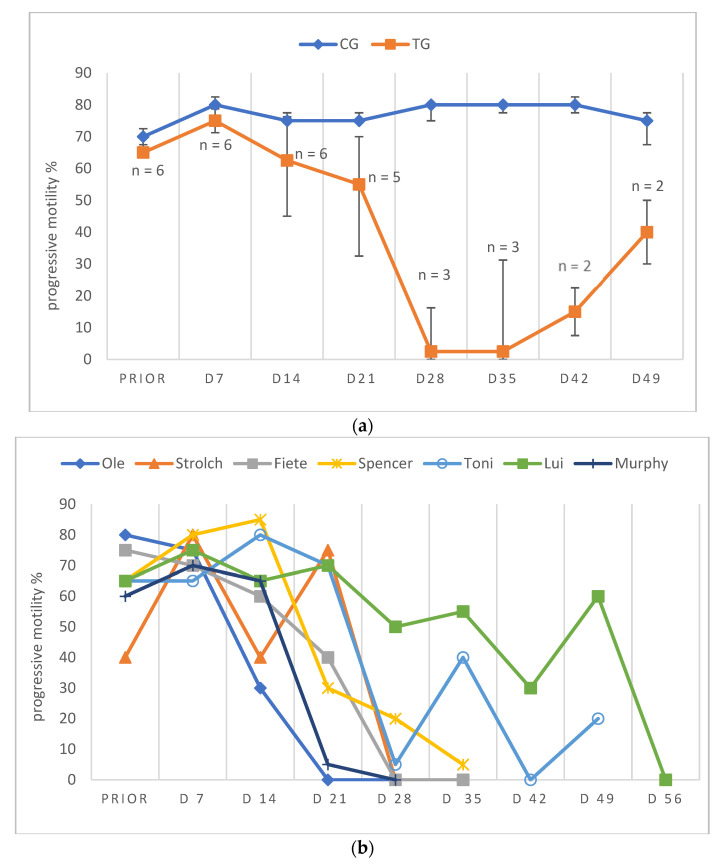
(**a**) Percentage of progressive motility given as median (Q1, Q3) over time of animals treated with a 4.7 mg deslorelin implant (TG, n = 6, if not indicated otherwise) and saline-treated control animals (CG, n = 3). “Prior” indicates “prior to treatment”. (**b**) Percentage of progressive motility of spermatozoa given as individual courses of animals treated with a 4.7 mg deslorelin implant (TG, n = 7) over time. “Prior” indicates “prior to treatment”.

**Figure 6 animals-12-02379-f006:**
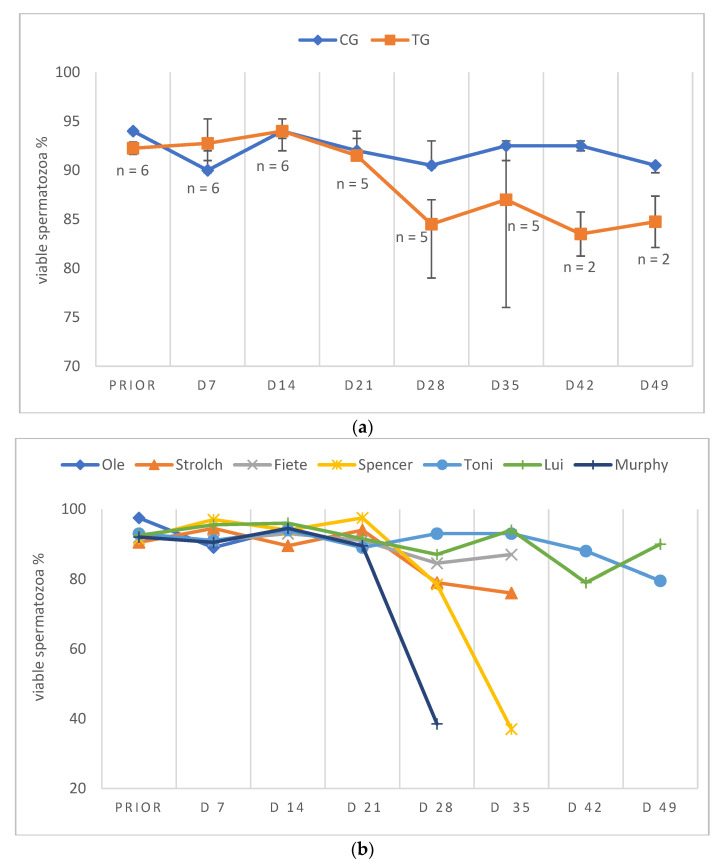
(**a**) Percentage of viable sperm given as median (Q1, Q3) over time of animals treated with a 4.7 mg deslorelin implant (TG, n = 6, if not indicated otherwise) and saline-treated control animals (CG, n = 3). “Prior” indicates “prior to treatment”. (**b**) Percentage of viable sperm given as individual courses of animals treated with a 4.7 mg deslorelin implant (TG, n = 7) over time. “Prior” indicates “prior to treatment”.

**Figure 7 animals-12-02379-f007:**
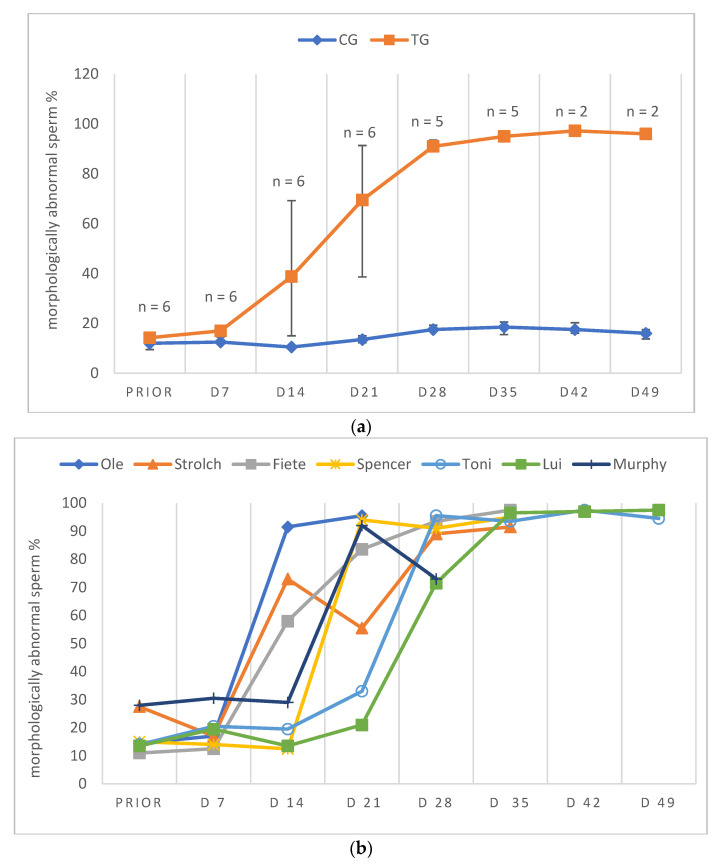
(**a**) Percentage of morphologically abnormal sperm given as median (Q1, Q3) over time of animals treated with a 4.7 mg deslorelin implant (TG, n = 6, if not indicated otherwise) and saline-treated control animals (CG, n = 3). “Prior” indicates “prior to treatment”. (**b**) Percentage of morphologically abnormal sperm given as individual courses of animals treated with a 4.7 mg deslorelin implant (TG, n = 7) over time. “Prior” indicates “prior to treatment”.

**Figure 8 animals-12-02379-f008:**
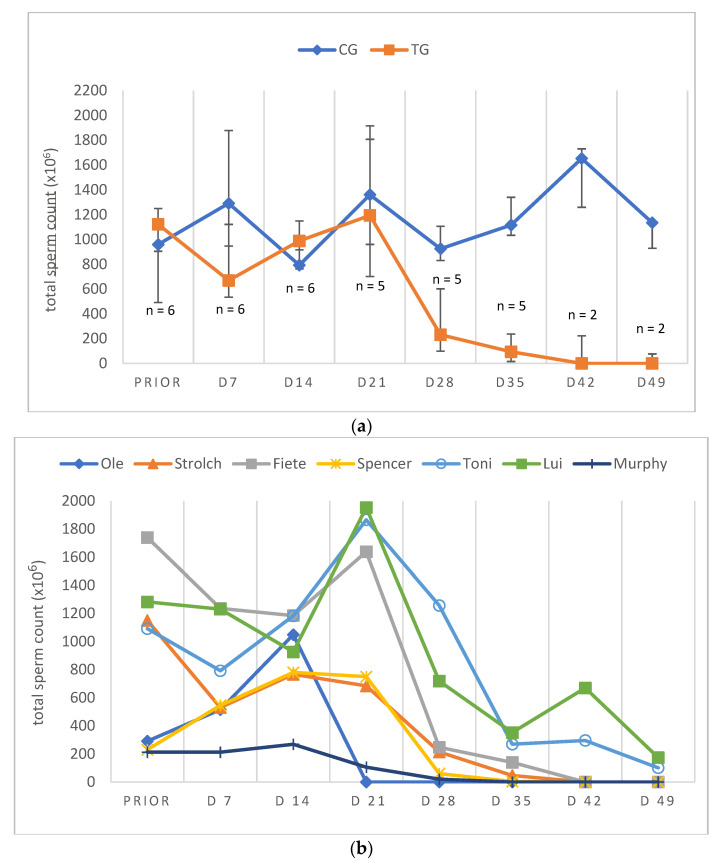
(**a**) Total sperm count (×10^6^) over time given as median (Q1, Q3) of animals treated with a 4.7 mg deslorelin implant (TG = 6, if not indicated otherwise) and saline-treated control animals (CG, n = 3). “Prior” indicates “prior to treatment”. (**b**) Total sperm count (×10^6^) given as individual courses of animals treated with a 4.7 mg deslorelin implant (TG, n = 7). “Prior” indicates “prior to treatment”.

**Figure 9 animals-12-02379-f009:**
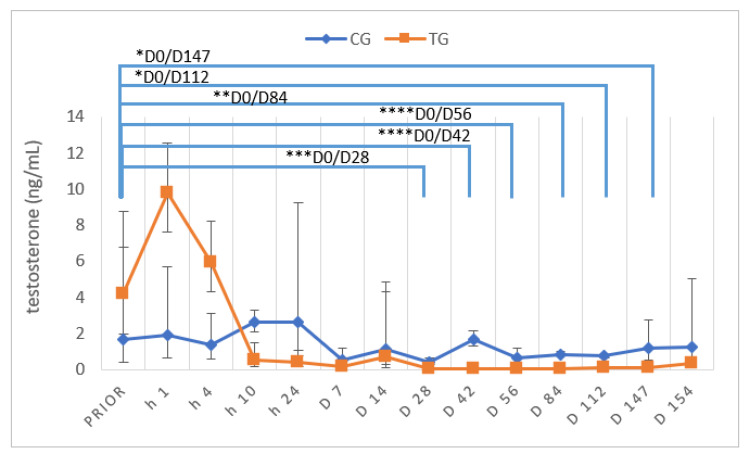
Plasma testosterone concentrations over time in ng/mL; given as geometric mean ($\bar{x_{g}}$) and scatter range whose limits are defined by $\bar{x_{g}}$ × dispersion factor^±1^; * *p* < 0.05, ** *p* < 0.01, *** *p* < 0.001, *****p* < 0.0001; animals treated with a 4.7 mg deslorelin implant (TG, n = 6) and saline-treated control animals (CG, n = 3). “Prior” indicates “prior to treatment”.

**Figure 10 animals-12-02379-f010:**
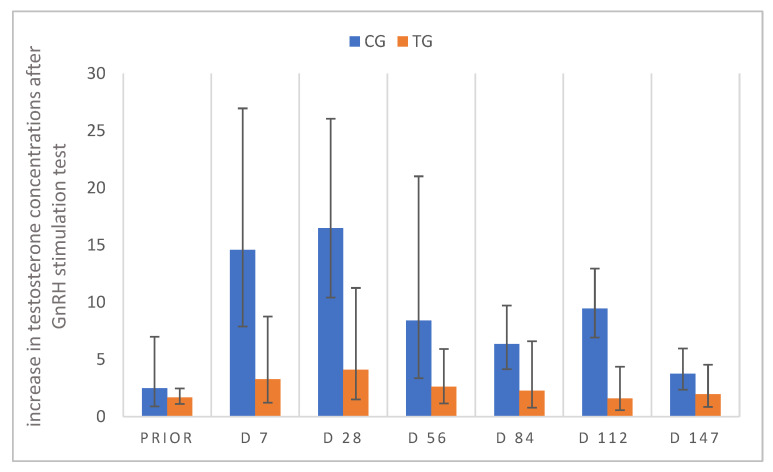
Relative increase in plasma testosterone concentrations 60 min after GnRH stimulation; testosterone values before stimulation are considered as 1, relative increase from pre- to post- stimulation is presented for study day PRIOR (day 14 prior to treatment), D7, D28, D56, D84, D112, D147; animals treated with 4.7 mg deslorelin implant (TG, n = 7); saline-treated control animals (CG, n = 3). Results are given as geometric mean $\bar{x_{g}}$ (DF). “Prior” indicates “prior to treatment”.

**Figure 11 animals-12-02379-f011:**
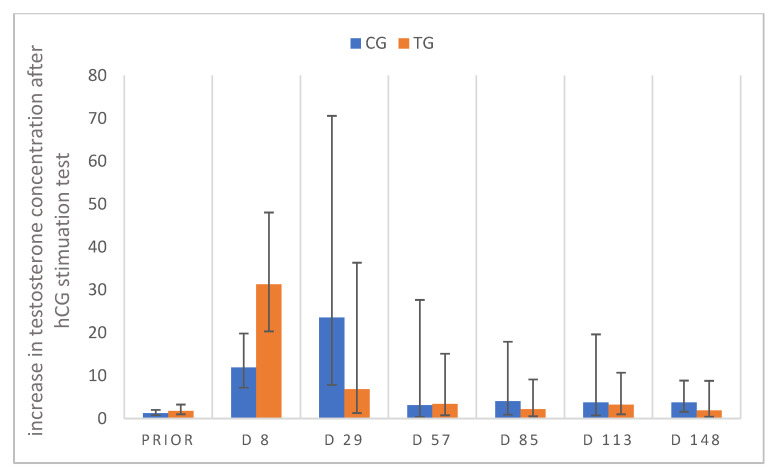
Relative increase in plasma testosterone concentrations 60 min after hCG stimulation; testosterone values pre-stimulation are considered as 1, relative increase from pre- to post-stimulation is presented for study day PRIOR (day 11 prior to treatment), D8, D29, D57, D85, D113, D148 animals treated with a 4.7 mg deslorelin implant (TG, n = 7) and saline-treated control animals (CG, n = 3). Results are given as geometric mean $\bar{x_{g}}$ (DF). “Prior” indicates “prior to treatment”.

**Figure 12 animals-12-02379-f012:**
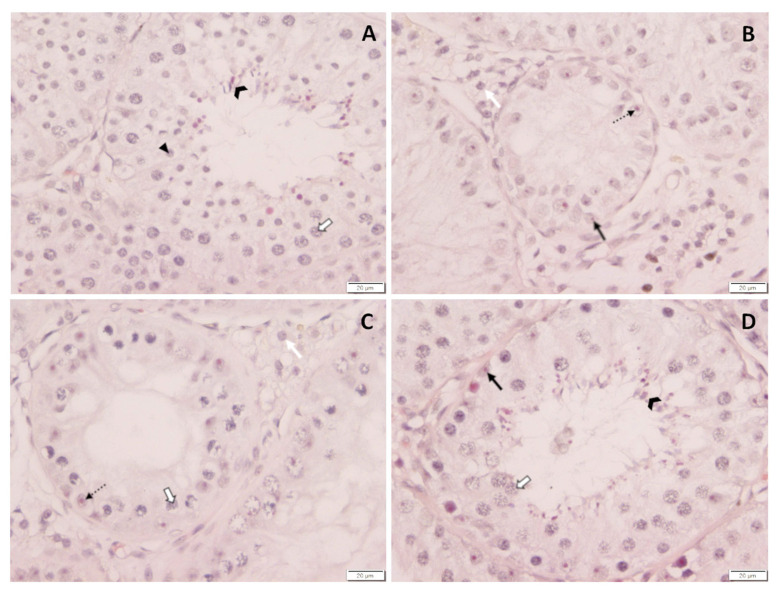
(**A**–**D**): Histological evaluation of testicular tissue of (**A**) saline-treated control dog and (**B**–**D**) deslorelin-treated dogs. (**A**) Normal spermatogenesis. (**B**) Arrest of spermatogenesis at the level of spermatogonia. (**C**) Arrest of spermatogenesis at the level of spermatocytes. (**D**) Reversed effects of treatment as indicated by normal spermatogenesis and spermatozoa in the semen sample on D147 after treatment; 
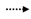
 spermatogonia; 
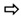
 spermatocytes; 
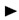
 round spermatids; 
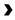
 elongated spermatids; 
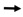
 Sertoli cells; white arrow, Leydig cells.

**Figure 13 animals-12-02379-f013:**
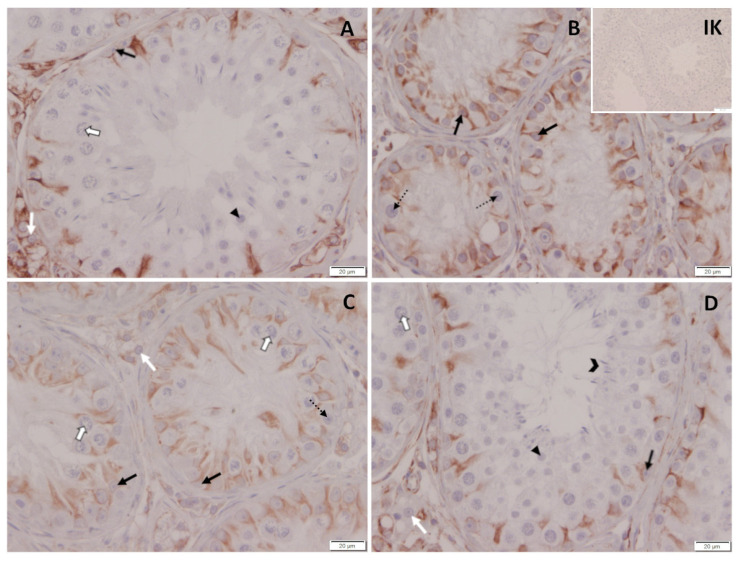
Immunopositive staining of testicular tissues against Vimentin. (**A**) Saline-treated control dog and (**B**–**D**) deslorelin-treated dogs. (**A**) Normal spermatogenesis. (**B**) Arrest of spermatogenesis at the level of spermatogonia. (**C**) Arrest of spermatogenesis at the level of spermatocytes. (**D**) Reversed effects of treatment as indicated by normal spermatogenesis and spermatozoa in the semen sample on D147 after treatment; IK: isotype control; 
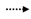
 spermatogonia; 
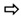
 spermatocytes; 
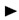
 round spermatids; 
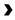
 elongated spermatids; 
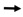
 Sertoli cells; white arrow, Leydig cells.

**Figure 14 animals-12-02379-f014:**
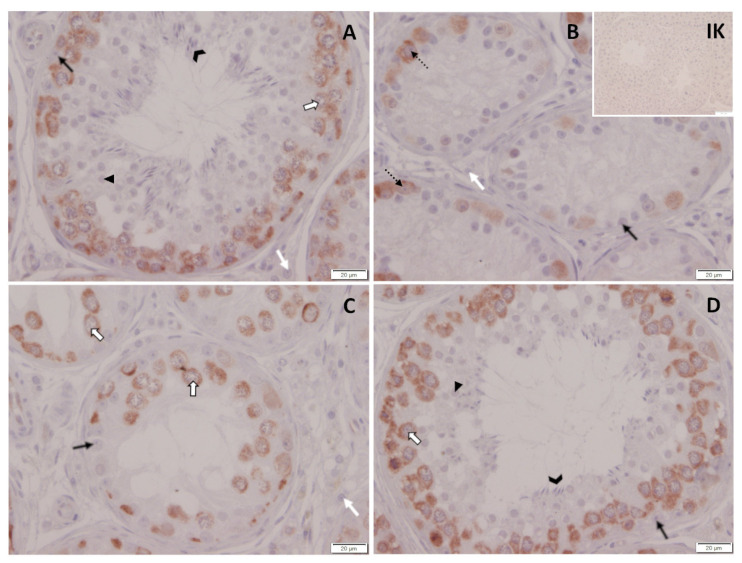
Immunopositive staining of testicular tissues against DAZL. (**A**) Saline-treated control dog and (**B**–**D**) deslorelin-treated dogs. (**A**) Normal spermatogenesis. (**B**) Arrest of spermatogenesis at the level of spermatogonia. (**C**) Arrest of spermatogenesis at the level of spermatocytes. (**D**) Reversed effects of treatment as indicated by normal spermatogenesis and spermatozoa in the semen sample on D147 after treatment; IK: isotype control; 
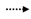
 spermatogonia; 
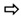
 spermatocytes; 
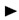
 round spermatids; 
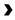
 elongated spermatids; 
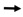
 Sertoli cells; white arrow, Leydig cells.

**Table 1 animals-12-02379-t001:** Endocrine (testosterone, in ng/mL) and histological findings related to spermatogenesis as well as classification of semen findings in individual dogs of the control and deslorelin-treated group on the day of hemicastration.

Groups	Testosterone (ng/mL)	Most Developed Germ Cell:	Level of Arrest of Spermatogenesis (Most Developed)	Semen Findings
Control group:				
Piet Knopf Merlin	0.25 1.91 3.78	Elongated spermatids	Full/Normal spermatogenesis	Normospermia
Treated group:				
Ole	0.10	Spermatogonia	Arrest at level of spermatogonia	Aspermia (no ejaculate)
Strolch	0.10	Spermatogonia		
Fiete	0.16	Spermatogonia		
Spencer	0.13	Spermatogonia		
Murphy	0.27	Spermatogonia		
Toni	1.76	Spermatocytes	Arrest at level of spermatocytes	Azoospermia (VOL: 0.1 mL)
Lui	3.16	Elongating spermatids	Full spermatogenesis	Pathospermia (VOL: 0.1 mL PM: 20% MAS: 97% *)

Underline: This is the headline for the group and the dog names are belonging to the respective group. * other parameters could not be evaluated due to low ejaculate volume.

## Data Availability

The data presented in this study are available on request from the corresponding author.

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
