# Peer review of "What Happens in Male Dogs after Treatment with a 4.7 mg Deslorelin Implant? I. Flare up and Downregulation"

_animals, 2022, doi:10.3390/ani12182379_

Round 1

Reviewer 1 Report

This paper is a study on the morphological changes of the testes and scrotum, which are the gonads, and the amount of ejaculated semen when Deslorelin implant, a GnRH agonist, is long-acting implanted to male dogs. Although it is well known how the GnRH agonist affects the male and female reproductive system in dogs, in this paper, histological examination and immunostaining of the seminiferous tubules were also attempted, providing a lot of information to the veterinarians.

And it is considered to be a good paper that helps a lot in applying the GnRH agonist implant, which is an alternative, to veterinary clinical practice in the reality that castration is often reluctant in terms of animal welfare.

A lot of research has been done and the English is well written, so it is considered acceptable as it is.

Author Response

Thank you very much for your kind words and the nice and positive review of our work!

Sincerely

Sandra

Reviewer 2 Report

The paper documents in great detail the events following deslorelin treatment in adult male dogs.  This is great information. In general, the paper can be made more succinct (in particular the discussion and the grammar needs improvement, it seems to repeat much of the results.  I have made some specific suggestions, but it would benefit from native English speaker input.  In particular verb tenses and preposition.

15 figures (some panels) and three large tables is a lot- and I don’t think all 15 figures are necessary- can some of these be summarized in the text or combined.  if the authors decide to keep all the figures- format them all similarly- for example since most of the data are longitudinal, the graphs can all be lines rather than bars

The paper describes a study done to fill in some of the gaps in knowledge about the effects of deslorelin on the male reproductive function

If the implant is designed to last 6 months, why were only 5 months evaluated?

Report data in the same format/ units

Line 17 while the message is conveyed, the wording is a bit awkward in the simple summary- the wording in the abstract is clearer.

 Basal testosterone only the last dog is reported whereas the azoospermic ejaculates are reported for the earliest and latest dog to reach

line 19-20 mixes 5 months with D147

Small grammatical errors

Line 38- interest in, not on

Line 65 SRI needs to be spelled out the first time it is used

Line 83- should read human chorionic gonadotropin

Figure 1 is very informative, but as the figures should stand on their own (independent of text), including what TG and CG mean would be good

Line 163-164 intra and inter-assay coefficients.. of variation?

Line 194 it should be location (A particular point or place in physical space) rather than localization (the act of localizing).

It is great to see the care that was taken to appropriately analyze the data (normal vs non-normal distribution)

Line 278-“ two dogs each”  rephrase it is not clear what each refers to.

Lines 276-281 also not clear it is difficult to follow what happened to each dog.  There are 7 in the TG, 2 had not clinical signs of inflammation (that leaves 5) then the next sentence says two each- does that leave one?- then “one of these”- which?  “swelling also occurred on D2 in the other one? (the last one that was left after the “two each”? Or one of the two sets of two?)

Figure 2- the graph depicts the TG (n=7), why do some columns have 9 animals and some 6? Shouldn’t there be 7 animals in each column?

Line 324- consider using heterogeneous rather than inhomogeneous

Line 342 Different from this or in contrast to this

Line 348-349 remained at this level

Because azoospermia and aspermia are easily confused, it might improve clarity if you include definitions when first used)

While including the graphs for the individual as well as the combined values shows the variation among individuals – it is probably not necessary to do this for every parameter- the Q1-Q3 interval also provides a measure of this variation.

Is there a reason not to plot the testosterone concentration with the same type of graph as all the others?

439 reword this – the authors use before to refer to prior to treatment and then also to refer to prestimulation- Using different words improves clarity, for example pre-stimulation compared to post-stimulation to differentiate from before and after treatment-

Alternatively “before treatment” can also be referred to as D -7 or however long before treatment that was, that way the terminology, with D0 the day of treatment.

Table 1- indicate in the legend that it is plasma testosterone concentrations, spell out DF why was SD not used?  (Fig 12 explains that this is a geometric mean- this needs to be added to all for consistency

Line 471- this sentence sounds truncated: “…, but on every other treatment day.”

How do the authors explain the low pre-stimulation plasma testosterone concentrations in the control group?

498- Spermatogenesis was arrested at the level, rather than “on the level”- correct throughout the manuscript

Table 3 title-it basically repeats all the headings, can you summarize to something like Spermatogenesis and endocrine findings in dogs treated with deslorelin and controls?

Histology- figures would be better if the white balance is adjusted as the contrast would make the features clearer- legend is great

537- Since  one datapoint cannot vary “… intensity differed between treated dog that had reversed and the CG:, would be better than “varied”

565 “our study is currently the first describing” should read  “our study is the first to describe”

567  correct grammar “Currently only two studies have investigated”

Comments about behavior (libido, pelvic thrusts) are not mentioned until the conclusions- If this was part of the observations this should be mentioned in the M&M and the results

Author Response

Dear reviewer, dear editor, 

please find attached the comments

Sincerely

Sandra

Round 2

Reviewer 2 Report

The authors have done a great job with improvements, I am glad to see you opted to remove the tables and keep the graphs.  The only remaining comment i have is that the histology images still seem to have a grey rather than white backgound which takes away from the quality of the images.  

Very nice study.

Author Response

Thank you very much for your nice review. I am not sure why the figures seem to have a grey background. We checked them on various computers and cannot see it. I am very sorry, but don't know how to improve it?

Best

Sandra